# Non-Covalent Linkage of Helper Functions to Dumbbell-Shaped DNA Vectors for Targeted Delivery

**DOI:** 10.3390/pharmaceutics15020370

**Published:** 2023-01-21

**Authors:** Pei She Loh, Volker Patzel

**Affiliations:** 1Healthy Longevity Translational Research Programme, Yong Loo Lin School of Medicine, National University of Singapore, Singapore 117597, Singapore; 2Department of Microbiology & Immunology, Yong Loo Lin School of Medicine, National University of Singapore, Block MD4, Level 5, 5 Science Drive 2, Singapore 117597, Singapore; 3Department of Medicine, Addenbrooke’s Hospital, University of Cambridge, Cambridge CB2 0QQ, UK

**Keywords:** dumbbell-shaped DNA vectors, targeted delivery, GalNac3, aptamers, suicide gene therapy, cleavable linker

## Abstract

Covalently closed dumbbell-shaped DNA delivery vectors comprising the double-stranded gene(s) of interest and single-stranded hairpin loops on both ends represent a safe, stable and efficacious alternative to viral and other non-viral DNA-based vector systems. As opposed to plasmids and DNA minicircles, dumbbells can be conjugated via the loops with helper functions for targeted delivery or imaging. Here, we investigated the non-covalent linkage of tri-antennary N-acetylgalactosamine (GalNAc3) or a homodimer of a CD137/4-1BB-binding aptamer (aptCD137-2) to extended dumbbell vector loops via complementary oligonucleotides for targeted delivery into hepatocytes or nasopharyngeal cancer cells. Enlarging the dumbbell loop size from 4 to 71 nucleotides for conjugation did not impair gene expression. GalNAc3 and aptCD137-2 residues were successfully attached to the extended dumbbell loop via complementary oligonucleotides. DNA and RNA oligonucleotide-based dumbbell-GalNAc3 conjugates were taken up from the cell culture medium by hepatoblastoma-derived human tissue culture cells (HepG2) with comparable efficiency. RNA oligonucleotide-linked conjugates triggered slightly higher levels of gene expression, presumably due to the RNaseH-mediated linker cleavage, the release of the dumbbell from the GalNAc3 residue and more efficient nuclear targeting of the unconjugated dumbbell DNA. The RNaseH-triggered RNA linker cleavage was confirmed in vitro. Finally, we featured dumbbell vectors expressing liver cancer cell-specific RNA *trans*-splicing-based suicide RNAs with GalNAc3 residues. Dumbbells conjugated with two GalNAc3 residues triggered significant levels of cell death when added to the cell culture medium. Dumbbell vector conjugates can be explored for targeted delivery and gene therapeutic applications.

## 1. Introduction

Various ground-breaking novel technologies including CRISPR/Cas genome editing, suicide gene therapy of cancer and virus infection, somatic cell reprogramming and genetic vaccination depend on non-viral, non-integrating transient gene expression vectors that are not silenced in primary cells and which vanish once their job is done [1]. Dumbbell-shaped DNA vectors or dumbbells composed of double-stranded linear expression cassettes and single-stranded hairpin loops on both ends meet the above requirements and have shown promising results in preclinical and clinical trials [2,3,4,5,6]. Dumbbell vectors combine several advantages which are only partly held by alternative non-viral genetic vectors such as plasmids, minicircles or chemically modified RNA: (1) the minimal size and hydrodynamic diameter of dumbbells facilitates cellular and nuclear delivery [7]; (2) no upper or lower size limitations have been reported for dumbbells so far; (3) dumbbells do not harbor free 5′ or 3′ ends, are completely exonuclease-resistant and were found to be more stable compared with plasmids or chemically modified RNA [7]; (4) dumbbells lack bacterial sequences and antibiotic resistance genes and are not immunotoxic and redosable; (5) dumbbells can be designed to lack non-transcribed extragenic sequences and do not suffer from transgene silencing in primary cells and in vivo; (6) they are non-integrating vectors and considered to be safe; and (7) via the single-stranded loops, dumbbells can be conjugated with helper functions for immune activation, imaging and targeted delivery. Targeted delivery, i.e., delivery of genetic vectors into the right target cells increasingly becomes a requirement for genetic therapies in order to improve activity and reduce off-target effects. Whereas many viral vectors exhibit some level of cell tropism, non-viral vectors depend on covalently or non-covalently linked helper functions including antibodies, aptamers, cell-penetrating peptides or other residues that recognize target cell surface markers. Such helper functions are frequently linked to liposomes and other nano-carriers that are complexing the genetic vectors, but it is more challenging to directly attach them to the genetic material itself. 

Small interfering RNA (siRNA) and small hairpin RNA (shRNA) were conjugated with tri-antennary N-acetylgalactosamine (GalNAc3) residues for targeted delivery into hepatocytes and the first siRNA-GalNAc3 conjugates received drug approval status recently [8]. The GalNAc3-mediated delivery is based on the interaction between the GalNAc3 residues and the asialoglycoprotein receptor (ASGPR) which is highly expressed on the surface of hepatocytes and several human cancer cell lines. Similarly, PSMA- and HIV-1 gp120-targeting aptamers were conjugated with RNAi effectors and uptake by human cells in vitro was successfully mediated [9,10]. The first time, we non-covalently attached GalNAc3 residues or a homodimer of a CD137-binding aptamer (aptCD137-2) to dumbbell vectors via the loops using complementary oligonucleotides [11,12]. MaxGFP protein or Herpes simplex virus thymidine kinase (HSVtk)-encoding dumbbell-GalNAc3 conjugates were taken up by hepatoblastoma-derived human tissue culture cells triggering MaxGFP expression, or cell death upon treatment with ganciclovir (GCV). In summary, dumbbell conjugates represent minimalistic DNA-based expression vectors that can be explored for smart delivery. 

## 2. Material and Methods

### 2.1. Plasmid Construction

Plasmid pGFP was cloned by inserting the maxGFP gene (CMV promoter, cds, SV40 ployA site) of the pMAX-GFP (Lonza) vector into the *Nde*I and *Bbs*I restriction endonuclease cleavage sites of the pVAX1 vector (Addgene). The SV40 enhancer, which functions as a DNA nuclear localizing signal (dNLS), was inserted into the *Nde*I and *Bbs*I site upstream of the CMV promoter. The HSVtk positive control plasmid carried a codon-optimized HSV1 thymidine kinase coding sequence inserted into the pVAX1 plasmid under the control of the CMV promoter [13]. 

### 2.2. Oligonucleotides

GalNAc3-linked oligonucleotides GalNAc3-DNA 5′-GCTATAAGTGTGCATGAGAAC-GalNAc3-3′ and GalNAc-RNA 5′-dGdCUAUAAGUGUGCAUGAGAAC-GalNAc3-3′ were derived from Microsynth (Balgach, Switzerland). First two 5’ terminal nucleotides are deoxyribonucletides. Primers for the production of dumbbells were derived from IDT. These included universal pVAX1 reverse gap-primers, 5′-pAAGGTCTTTTGACCT/idSp/GAAGCCATAGAGCCCACCG-3′; Loop primer (for ELAN), 5′-CTAGCGACCAGTTTTATTTTATTTTATTTTAGTTCTCATGCACACTTATAGCGGTTTGGTTTGGTTTGGTAACTGGTCG-3′; gap-primer for one GalNAc3, 5′-/5Phos/ATCCAGTTTTATTTTATTTTATTTTAGTTCTCATGCACACTTATAGCGGTTTGGTTTGGTTTGGTAACTGGA/idSp/ GCGATGTACGGGCCAGATATA-3′; gap-primer for two GalNAc3, 5′-/5Phos/ATCCAGTTTTATTTTATTTTATTTTAGTTCTCATGCACACTTATAGCGGGAAACCCGTTCTCATGCACACTTATAGCGGTTTGGTTTGGTTTGGTTTCTGGA/idSp/GCGATGTACGGGCCAGATATA-3′.

For RT-qPCR, Fw_maxGFP (5′-ATCGAGTGCCGCATCACC-3′) and Rv_maxGFP (5′-ACTCATCGAGCTCGAGATCTGG-3′) were used. Fw-Beta actin (5′-CTGGCACCCAGCACAATG-3′) and RP-beta actin (5′-GCCGATCCACACGGAGTACT-3′) were used as housekeeping genes. 

### 2.3. Dumbbell Vector Production

Dumbbell vectors harboring one terminal mismatch and one conjugation loop were manufactured using a protocol that combines the enzymatic ligation assisted by nucleases (ELAN) method and the gap-primer PCR (gpPCR) method [14,15]. Therefore, genes of interest were PCR amplified using a linearized DNA template and a mixture of the Q5 and Taq DNA polymerases involving one 5′ phosphorylated gap-primer and one conventional primer providing a *Xba*I cleavage site. The PCR product was cleaved with *Mlu*I, analyzed using agarose gel electrophoresis and purified using phenol/chloroform/isoamylalcohol (PCI) extraction, chloroform/isoamylalcohol (CI) re-extraction and ethanol precipitation. Conjugation loops were 5′ phosphorylated with T4 polynucleotide kinase (PNK) and ligated to the *Mlu*I-cleaved PCR product overnight at 22 °C using T4 DNA ligase in the presence of *Mlu*I and *Sgs*I enzymes. Concurrently, the hairpin loop resulting from the gpPCR was ligated this way. Non-covalently closed by-products and primers were removed by T7 DNA polymerase. The resulting dumbbells were purified using a PCR purification kit (Qiagen, Hilden, Germany). The resulting dumbbells were proven to be exonuclease resistant (Appendix A).

Dumbbell vectors harboring two terminal mismatches were manufactured using the gpPCR method [13]. Therefore, genes of interest were PCR amplified using a linearized DNA template and a mixture of the Q5 and Taq DNA polymerases involving two 5′ phosphorylated gap-primers. The resulting gpPCR product was ligated overnight at 22 °C using T4 DNA ligase. Non-covalently closed by-products and primers were removed by T7 DNA polymerase. The resulting dumbbells were purified using a PCR purification kit (Qiagen).

## 3. Formation of Dumbbell Conjugates

### 3.1. Dumbbell-GalNAc3-Conjugates 

A total of 3.5 pmol GalNAc3-DNA or GalNAc3-RNA oligonucleotide was annealed with 3.5 pmol dumbbell DNA in 20 μL 10× hybridization buffer (1 M NaCl, 0.1 M MgCl2, 200 mM Tris-HCl, pH 7.4) in the presence of 20% *v*/*v* of PEG4000. The solution was denatured at 80 °C for 5 min and then incubated at 37 °C for 1 h. The resulting dumbbell-GalNAc3 conjugates were cleaved with *Ase*I (Thermo Fisher, Waltham, MA, USA) and GalNAc3 attachment to the conjugation loops was monitored in 1.5% agarose gel shift assays. Dumbbell-conjugates were purified using Sephadex gel permeation chromatography and ethanol precipitation.

### 3.2. Dumbbell-aptCD137-2-Conjugates

A total of 3.5 pmol of the aptCD137-2 homodimer was annealed with 3.5 pmol dumbbell DNA as described above. 

## 4. Tissue Cell Culture

Human cell lines HEK293T and HepG2 were purchased from ATCC. Human HEK293T and HepG2 cells were cultured in Dulbecco’s Modified Eagle’s Medium (DMEM, ThermoFisher Scientific) supplemented with 10% *v*/*v* heat-inactivated Fetal Bovine Serum (Hyclone) and 1% penicillin–streptomycin solution (Invitrogen) at 37 °C in a humidified incubator with 5% CO_2_. The cells were passaged every 3–4 days.

## 5. Uptake of Dumbbell-GalNAc3-Conjugates from the Tissue Cell Culture Medium

### 5.1. Uptake of MaxGFP Dumbbell-GalNAc3-Conjugates

HepG2 cells were trypsinized, washed with 10 mL DMEM, 0.05 × 10^6^ cells were resuspended in 30 μL DMEM and 3.5 pmol MaxGFP dumbbell-GalNAc3-conjugates dissolved in 20 µl of water were added and incubated with the cells for 4 h before seeding them again. By adding the dumbbell conjugates to HepG2 cells in suspension, we achieved a higher concentration of dumbbell conjugates in the medium to observe a stronger expression of MaxGFP. 

### 5.2. Uptake of HSVtk Dumbbell-GalNAc3 or -2GalNAc3-Conjugates

0.05 × 10^6^ HepG2 cells were seeded in 24 wells and 0.35 pmol HSVtk dumbbell-GalNAc3 or *-2GalNAc3-*conjugates dissolved in 20 µL of water were added after 24 h to the DMEM culture medium.

## 6. qPCR Quantification of Uptaken MaxGFP Dumbbell-GalNAc3 DNA

After 24 h of exposure, cells were harvested and episomal nucleic acids including uptaken dumbbell-conjugates were isolated using the RNeasy^®^ (Qiagen) Plus kit following the manufacturer’s protocol. For SYBR Green detection of dumbbell DNA, 1 μL of the episomal nucleic acid sample was mixed with 1X SYBR^®^ Select Master Mix (ThermoFisher) for CFX and each 0.5 µM forward and reverse primers in a 10 µL reaction. All reactions were run in duplicate. The dumbbell vector DNA was quantified using absolute qPCR quantification based on a standard curve created with dumbbell vector DNA.

## 7. Lipofection of HepG2 Cells

HepG2 cells were transfected with plasmids or dumbbell-shaped DNA minimal vectors using Lipofectamine^®^ 3000 (ThermoFisher) following the manufacturer’s protocol. In short, 500 ng of DNA and 1 μL of P3000 were diluted in 25 μL of Opti-MEM and then mixed with 1.5 μL Lipofectamine 3000 (diluted in 25 μL of Opti-MEM). The mixture was incubated at room temperature for 5 min before adding onto HepG2 cells.

## 8. Flow Cytometry

For FACS analysis, the media was aspirated, and the cells were rinsed once with PBS before trypsinization with 200 μL of 1X trypsin-EDTA. The trypsinized cells were collected by centrifugation at 4200 rpm for 6 min in 1 mL of media. The pelleted cells were resuspended in 500 μL of 1X PBS. FACS was performed with 10,000 cells for the NTC and >5000 cells per sample using an LSRFortessa cell analyzer, and FACSDiva software v6.1.3 was used for the acquisition of the samples. FlowJo software V7.6.1 was used for data analysis.

## 9. Preparation of Ganciclovir Working Stock

GCV was purchased from Sigma (Darmstadt, Germany) as a 100 mg ready-to-mix powder form. To dissolve the powder for master stock for cell culture experiments, 10 mg of GCV was dissolved in 1 mL of 0.1 N/0.1 M HCl (final concentration: 10 mg/mL). For a 10 mM working stock (for conditions ranging from 1–100 µM), 255 µL of 10 mg/mL GCV master stock was diluted in 745 µL of 0.1 N HCl. For 1 mM working stock (for conditions ranging from 0.1 µM and 0.3 µM), 100 µL of 10 mM working stock was further diluted in 900 µL of 0.1 N HCl.

## 10. AlamarBlue^®^ Cell Viability Assay

To trigger the death of dumbbell-conjugate transfected HepG2 cells, GCV was added to the cells at a concentration of 100 μM, 24 h post-transfection. The cell death activity was monitored using the AlamarBlue^®^ (ThermoFisher) cell viability reagent every 24 h for 6 days with addition of fresh media and the drug every day. The fluorescence was measured at 530 nm/590 nm after 90 min of incubation.

### Statistical Analysis 

Error bars represent standard errors of the arithmetic mean (±SEM) of three independent experiments. An unpaired student’s *t*-test was used to determine significance when comparing the two groups. GraphPad Prism 9 software was used for the statistical analysis. *p* values are as indicated on the graphs.

## 11. Results

### Increasing the Size of One Dumbbell Vector Loop Does Not Impair Gene Expression

Dumbbell-shaped DNA vectors are unique in the way they combine double-stranded expression cassettes with single-stranded loops. We explored the non-covalent attachment of residues for targeted delivery to the dumbbell loops via complementary base pairing. The targeting residues were covalently linked to antisense oligonucleotides (DNA or RNA) with complementarity to one of the dumbbell loops which we refer as the conjugation-loop in the following. To facilitate conjugation, the conjugation-loop size was enlarged from 4 nt, the standard loop size, to 41 or 71 nt using sequences that were predicted not to fold (or as little as possible) an intrinsic DNA secondary structure (Appendix A). HEK293T cells were then transfected with MaxGFP-expressing dumbbells featured with 4 nt, 41 nt or 71 nt conjugation-loops and MaxGFP expression was quantified (Figure 1). The increasing conjugation-loop size did not impair gene expression indicating that larger loops do not compromise nuclear vector diffusion. Notably, dumbbell vectors produced using gap-primer PCR or a combination of gap-primer PCR and the enzymatic ligation assisted by nucleases (ELAN) method harboring mismatches in one or both terminal stem-loop structures triggered significantly higher levels of gene expression compared to perfectly base-paired dumbbells and at a level that was comparable with the expression triggered by a plasmid [14,15]. We reported previously that such designed dumbbells exhibit facilitated nuclear diffusion rates [14]. As larger conjugation-loops are expected to facilitate the binding of the antisense DNA or RNA oligonucleotides and the formation of a resulting B- or H-form helix, we decided to proceed with 57 nt conjugation-loops for residue conjugation.

## 12. Non-Covalent Conjugation of Dumbbell Vector DNA with GalNAc3 and aptCD137-2 Residues via Complementary Base Pairing

The conjugation-loops were designed to be completely unstructured or to exhibit as little as possible internal secondary structure formation and to harbor a central 21 nt antisense oligonucleotide binding domain with two adjunct 18 nt spacers, one on each side (Appendix A, Figure 2A). To non-covalently attach the GalNAc3 and aptCD137-2 residues to the dumbbells’ conjugation loops, the residues were covalently linked to loop-binding antisense oligonucleotides (Figure 2B). GalNAc3 labeled DNA and RNA oligonucleotides were generated by chemical synthesis. The two aptamer domains of the aptCD137-2 homodimer were bridged via a RNA linker and the aptCD137-2 sequence was generated using in vitro transcription. The GalNAc3 and aptCD137-2 labeled oligonucleotides were then annealed to the conjugation-loops of HSVtk and/or MaxGFP-expressing dumbbell vectors (Figure 2B). The molecular weights of the MaxGFP- and HSVtk-expressing dumbbells were both with 1.3 × 10^6^ g/mol relatively large compared with the molecular weight of the GalNAc3 labelled oligonucleotides, exhibiting molecular weights of 8.4 × 10^3^. To monitor the successful conjugation using electrophoretic mobility shift assays, the dumbbell-GalNAc3- but not the dumbbell-aptCD137-2-conjugates were cleaved using the *Ase*I restriction endonuclease before analyzing the resulting fragments on agarose gels (Figure 2C,D). The use of stoichiometric oligonucleotide to dumbbell ratios for annealing resulted in incomplete GalNAc3 conjugation of the dumbbell vector DNA; however, virtually 100% conjugation efficiency was achieved when the RNA- or DNA-GalNAc3 oligonucleotides were used in 2- or 10-fold molar excess (Figure 2C). In the case of the aptCD137-2 aptamer conjugation, an equal molar amount or a 2-fold molar excess of the aptamer yielded a conjugation efficiency of 80% or 81%, respectively (Figure 2D).

## 13. GalNAc3-RNA but Not -DNA Linkers Are Cleavable by RNaseH

RNA in heteroduplexes formed between complementary RNA and DNA can be cleaved by endogenous RNaseH. As a consequence, dumbbell RNA but not DNA linker-conjugates can release the GalNAc3 residue from the dumbbell after delivery into the cytoplasm to facilitate diffusion of the dumbbell through the nuclear pore complex. We investigated the cleavability of our dumbbell-GalNAc3 conjugates by RNaseH using an in vitro assay. Therefore, the conjugates were exposed to RNaseH for 120 min. To better visualize the release of the small GalNAc3 residue from the relatively large dumbbell vector DNA using a gel shift assay, the conjugation loops were cleaved off with *Ase*I before loading the samples on a gel. As expected, the gel shift assay indicates cleavability and release of the GalNAc3 residue from the RNA- but not the DNA-linker dumbbell conjugate (Figure 2E).

## 14. Dumbbell-GalNAc3 Conjugates Are Taken up by Hepatoblastoma-Derived Human Tissue Culture Cells Triggering MaxGFP Expression

MaxGFP Dumbbell-GalNAc3 conjugates were mixed with trypsinized and pelleted HepG2 cells for 4 h before seeding the cells and after 48 h we quantified the levels of uptaken dumbbell vector DNA using qPCR (Figure 3B). Our data show that dumbbell-DNA- and dumbbell-RNA-GalNAc3 conjugates were taken up by HepG2 cells with comparable efficiency (Figure 3B). However, the uptake of dumbbell-GalNAc3 conjugates from the medium was significantly less efficient compared with delivery via lipofection. As dumbbell DNA might be adsorbed at the cell surface or stay unproductive in endosomes, we also quantified the transcribed MaxGFP mRNA using RT-qPCR (Figure 3C). Slightly more maxGFP mRNA was detected for the dumbbell-RNA as compared with the dumbbell-DNA conjugates, but the difference was not significant. The signals indicating the presence of db vector DNA or maxGFP mRNA in the NTCs are presumably a background signal which is either due to the presence of a minor contamination of the RT-/qPCR or an unspecific signal originating from the SYBR Green-based quantification protocol. To further improve the uptake and subsequent expression of dumbbell-GalNAc3 conjugates, two GalNAc3 residues were conjugated via an extended 71 nt conjugation loop harboring two antisense oligonucleotide binding sites (Figure 3D). Single and double GalNAc3 dumbbell conjugates linked via RNA- and DNA-linkers, were then incubated with HepG2 cells as described above and after 48 h we quantified the levels of MaxGFP protein expression using flow cytometry (Appendix A). On average, dumbbell-RNA-GalNAc3 conjugates gave more MaxGFP-positive cells (49.8%) compared with dumbbell-DNA-GalNAc3 conjugates (28.8%) (Figure 3E). No difference in MaxGFP expression was observed comparing dumbbell single and double GalNAc3 conjugates.

## 15. HSVtk Expressing Dumbbells Featured with Two GalNAc3 Residues at One Conjugation Loop Triggered Death of Hepatoblastoma-Derived Human Tissue Culture Cells upon Ganciclovir Treatment

As our intention is to develop dumbbell-GalNAc3 conjugates for suicide gene therapy of hepatocellular carcinoma (HCC), we attached two GalNAc3 residues via RNA linker oligonucleotides to a HSVtk expressing dumbbell vector (Figure 4A). To test the cell death activity of the conjugates, the vectors were added to the culture medium of HepG2 cells. The HSVtk dumbbell-2-GalNAc3 double-conjugate but not the HSVtk dumbbell-1-GalNAc3 conjugate featured with only one GalNAc3 residue, triggered significant death of HepG2 cells, i.e., a 34.7% reduction of cell viability at day six in an alamarBlue cell viability assay, upon 100 µM GCV treatment (Figure 4B). For comparison, lipofection of HepG2 cells with an HSVtk expressing plasmid or double-conjugated dumbbell vector reduced the cell viability by 49.8% or 54.2%. These data also indicate that dumbbell conjugation with 2 GalNAc3 residues via an RNA linker did not impair gene expression as compared with the unconjugated dumbbell. Notably, no significant reduction of cell viability was observed in the absence of GCV treatment.

## 16. Discussion

Dumbbell-shaped DNA vectors increasingly raise attention as a promising versatile naked DNA-based delivery vector system for gene therapeutic applications and for genetic vaccination. As opposed to plasmids and DNA minicircles, dumbbells can be covalently or non-covalently conjugated with helper functions for imaging, immune sensing or targeted delivery via the single-stranded loops. The latter are formed by chemically synthesized oligodeoxyribonucleotides which may be chemically modified, and which can either be added by direct ligation or modeled from primers used in a PCR reaction [14,15,16,17]. Loop conjugation of helper functions is not expected to impair the transcriptional activity of dumbbell vectors but may affect cellular and nuclear targeting. We investigated non-covalent linkage of GalNAc3 and aptCD137-2 residues for targeted delivery into hepatocytes and nasopharyngeal cancer cells. Therefore, antisense oligonucleotides (DNA or RNA) were covalently attached to these residues which could then be annealed via complementary base pairing towards extended conjugation loops. Extension of the conjugation loops, though in the absence of residue conjugation, did not impair dumbbell vector-based gene expression (Figure 1), and it is reasonable to assume that it did not impair nuclear diffusion either. That might be explained by the design of the loops, which were selected to be incapable of forming internal secondary structures which would render the dumbbells bulkier and more difficult to diffuse through the nuclear pore complex. Both RNA and DNA linker oligonucleotides could successfully attach residues to the conjugation loop of the dumbbell. We observed some evidence that conjugation via RNA linkers was more efficient than conjugation via DNA linkers as a lower excess of the oligo over the dumbbell yielded more conjugates. This finding may be explained by the higher stability of RNA:DNA base pairs as compared with DNA:DNA base pairs which would facilitate the nucleation process provided the number of RNA nucleation sites is not reduced due to secondary structure formation. A reduction of nucleation sites can be excluded in our example as both the RNA and DNA conjugation oligonucleotides were selected to be rather unstructured (Appendix A). In addition, the RNA but not the DNA linker could be cleaved by RNaseH to decouple the GalNAc3 residue and the dumbbell vector (Figure 2E). The use of cleavable or stimuli-labile linkers is strongly advised if large residues are being attached to a genetic vector or effector molecule. In our example, the GalNAc3 residue was rather small compared with the size of the dumbbell vector. Nevertheless, the use of an RNaseH-cleavable RNA linker did increase the number of MaxGFP-positive cells on average, indicating the release of the dumbbell from the GalNAc3 residue and more efficient nuclear targeting of the unconjugated dumbbell DNA (Figure 3D and Figure 5). While siRNA-GalNAc3 conjugates represent the clinical standard for targeted delivery of siRNA into hepatocytes, conjugation of GalNAc3 residues or aptamers to gene expression vectors including dumbbell vectors for targeted delivery has not yet been reported. In this proof-of-concept study, we demonstrated that conjugation of GalNAc3 residues to a 2186 bp MaxGFP-expressing dumbbell vector can facilitate vector delivery into HepG2 cells resulting in 29 to 51% MaxGFP-positive cells as measured using flow cytometry analyses (Figure 3E). According to our calculations, which are based on the cellular uptake of single GalNAc3 conjugates, about 10 dumbbell-GalNAc3 conjugate complexes were delivered in these experiments on average per cell. In addition, a HSVtk-expressing dumbbell featured with two GalNAc3 residues at a single conjugation loop triggered 34.7% death of HepG2 after addition to the cell culture medium in the presence of 100 µM GCV. The equivalent dumbbell-conjugate featured with only one GalNac3 residue did not exhibit a significant effect. The observation that double GalNAc3 conjugates triggered more cell death compared with single GalNAc3 conjugates indicates that multiple GalNAc3 residues attached to a single dumbbell may facilitate binding towards multiple ASGPR receptors and subsequent cellular uptake (Figure 5). However, this effect was not observed with MaxGFP-expressing dumbbells and requires further investigation. 

These data indicate that GalNac3-mediated targeted delivery of gene expression vectors such as dumbbells works, and it works better if more than one GalNac3 residue is conjugated. In principle, dumbbell vectors can be conjugated with more than two GalNac3 residues per conjugation loop and both dumbbell loops can be explored as conjugation loops to further improve delivery into hepatocytes. Though the MaxGFP expression triggered by cellular uptake of dumbbell-GalNAc3 conjugates was readily detectable using flow cytometry, it was scarcely visible under the fluorescence microscope with only single cells showing bright fluorescence [18]. On the other hand, uptake of HSVtk dumbbells effectively killed the targeted cells. This observation that HSVtk-expressing suicide vectors trigger a stronger phenotype than MaxGFP expression vectors may be explained by any or both of the following reasons: 1. a smaller dumbbell DNA cargo load which may be sufficient to kill a cell might not yet efficiently stain it with MaxGFP for detection, or 2. cells which were not primarily targeted by the dumbbell-conjugates might have been killed by the bystander effect that has been reported for the HSVtk/GCV gene-directed enzyme prodrug system [19]. Notably, HepG2 cells express significantly less ASGPR on their surface as compared with primary hepatocytes. As a consequence, one would expect stronger uptake of dumbbell-GalNAc3-conjugates by primary hepatocytes ex vivo or in vivo. In summary, we demonstrated that dumbbell vectors can efficiently be conjugated with helper functions for targeted delivery via cleavable linkers. Such vectors expressing novel mitochondrial targeting sequences are currently being explored for mitochondrial gene therapy in our lab. Our liver cancer-targeting GalNac3-conjugated suicide vectors are being tested in patient-derived xenograft (PDX) nude and humanized mouse models of HCC. As opposed to LNPs which can also be conjugated with helper functions including GalNAc3, naked dumbbell-conjugates are significantly smaller and expected to exhibit facilitated diffusion rates in the extracellular matrix. As a consequence, dumbbell-conjugates may identify single cells including cancer cells or metastasis more efficiently providing a minimalistic vector system that can complement or replace existing viral and non-viral carriers for gene therapeutic applications.

## Figures and Tables

**Figure 1 pharmaceutics-15-00370-f001:**
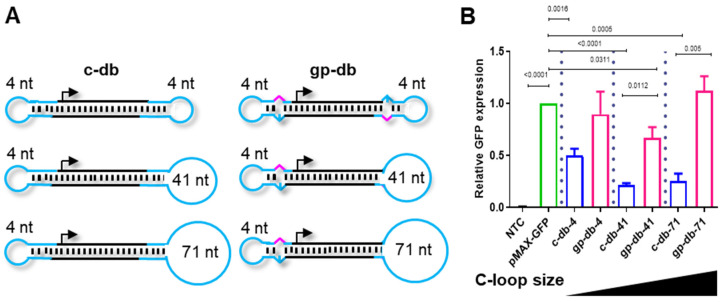
Increasing the size of conjugation loops (C-loop) does not impair the activity of dumbbell vectors generated using gap-primer PCR (gpPCR). (**A**) Design of conventional dumbbells (c-db) generated using the ELAN method only and of advanced dumbbells (gp-db) generated using gpPCR, each with various C-loop sizes. Black, double-stranded dumbbell core harboring the gene of interest; cyan, loops with indicated loop size; magenta, mismatch-triggering a basic position. (**B**) HepG2 cells were transfected with different dumbbell vectors harboring different C-loop sizes or the pMAX-GFP vector and MaxGFP expression was monitored 48 h post-transfection using flow cytometry. Mean ± SEM (*n* = 3). Significance was tested using the unpaired student’s *t*-test.

**Figure 2 pharmaceutics-15-00370-f002:**
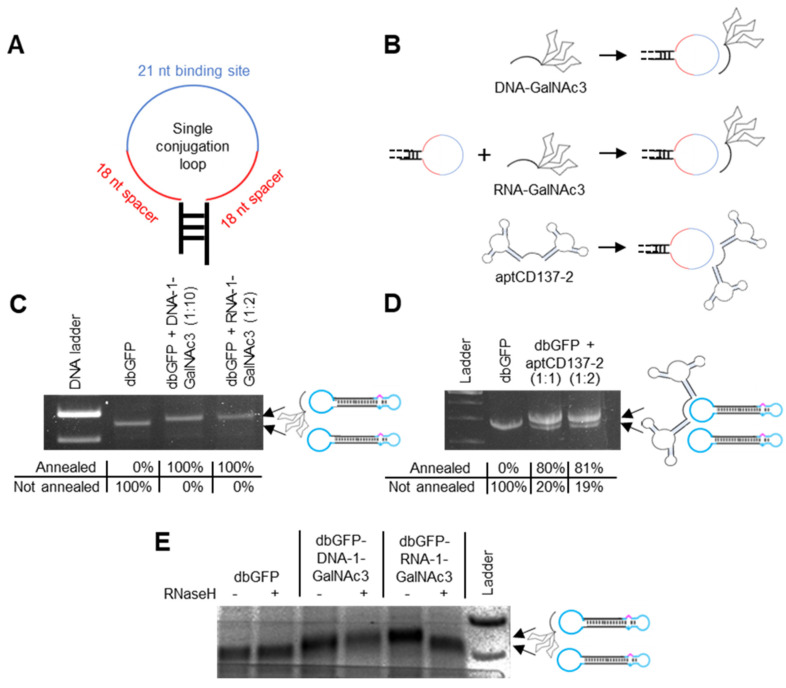
Non-covalent linkage of GalNAc3 and aptCD137-2 residues towards dumbbell vector conjugation loops via antisense DNA and RNA oligonucleotides. (**A**) Design of the single conjugation loop comprising a 21 nt conjugation oligonucleotide binding site (blue) and two flanking 18 nt spacers (red). (**B**) Schematic of attaching DNA-GalNAc3, RNA-GalNAc3 and aptCD137-2 residues to the dumbbell conjugation loop. (**C**) Agarose gel electrophoresis analyses of MaxGFP expressing dumbbell vectors (dbGFP) before and after annealing of a single DNA-1-GalNAc3 or RNA-1-GalNAc3 oligonucleotide at 10- (1:10) or 2-fold (1:2) molar excess. Notably, dumbbell-GalNAc3 conjugates were cleaved using the *Ase*I restriction endonuclease and the depicted section of the gel shows the extended conjugation loop only. The percentage of dumbbell-GalNAc3 conjugate formation as quantified using software imageJ version 1.53 u is indicated. (**D**) Agarose gel electrophoresis analyses of MaxGFP expressing dumbbell vectors (dbGFP) before and after annealing of a single aptCD137-2 homodimer using a stochiometric amount (1:1) or two-fold molar excess (1:2). The percentage of dumbbell-aptCD137-2 conjugate formation as quantified using software imageJ version 1.53 u is indicated. (**E**) Agarose gel electrophoresis analyses of dbGFP-DNA-1- and dbGFP-RNA-1-GalNAc3 conjugates after RNaseH cleavage.

**Figure 3 pharmaceutics-15-00370-f003:**
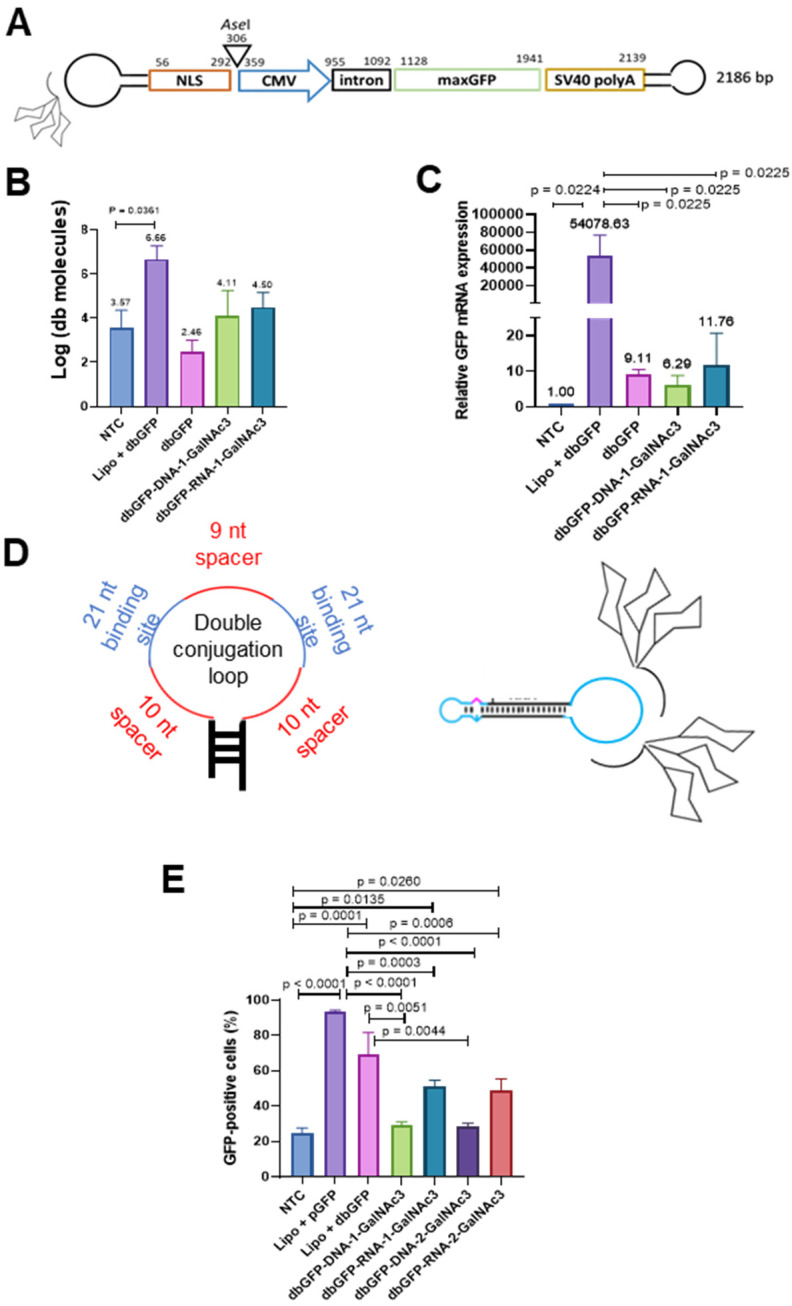
Uptake and expression of MaxGFP dumbbell-GalNAc3 conjugates by HepG2 cells from culture medium. (**A**) Design of the MaxGFP dumbbell-GalNAc conjugates. (**B**) HepG2 cells well exposed to DNA-linker- (dbGFP-DNA-1-GalNAc3), RNA-linker- (dbGFP-RNA-1-GalNAc3) or non-conjugated (dbGFP) MaxGFP dumbbells or transfected with dbGFP. Intracellular dumbbell DNA was isolated after 24 h and quantified using qPCR. (**C**) HepG2 cells well exposed to DNA-linker- (dbGFP-DNA-1-GalNAc3), RNA-linker- (dbGFP-RNA-1-GalNAc3) or non-conjugated (dbGFP) MaxGFP dumbbells or transfected with dbGFP. RNA was isolated after 24 h and quantified using RT-qPCR. (**D**) Design of the double conjugation loop comprising two 21 nt conjugation oligonucleotide binding sites (blue) flanked each by a 10 nt spacer (red) and separated by a 9 nt spacer (red), left panel; schematic of a dumbbell conjugate with two GalNAc3 residues attached to a single conjugation loop. (**E**) Average numbers of GFP = positive cells quantified using flow cytometry analyses of HepG2 cells exposed for 48 h to dbGFP-GalNAc3 conjugates added to the cell culture medium. NTC: No-transfection control; Lipo + pGFP: Cells transfected with pMaxGFP plasmid and Lipofectamine 3000; Lipo + dbGFP: Cells transfected with MaxGFP expressing dumbbell and Lipofectamine 3000; dbGFP-DNA-1-GalNAc3: MaxGFP expressing dumbbell conjugated with 1 GalNAc3 residue via a DNA linker; dbGFP-RNA-1-GalNAc3: MaxGFP expressing dumbbell conjugated with 1 GalNAc3 residue via a RNA linker: dbGFP-DNA-2-GalNAc3: MaxGFP expressing dumbbell conjugated with 2 GalNAc3 residues via a DNA linker: dbGFP-RNA-2-GalNAc3: MaxGFP expressing dumbbell conjugated with 2 GalNAc3 residues via a RNA linker. Mean ± SEM (*n* = 5). Significance was tested using the unpaired student’s *t*-test.

**Figure 4 pharmaceutics-15-00370-f004:**
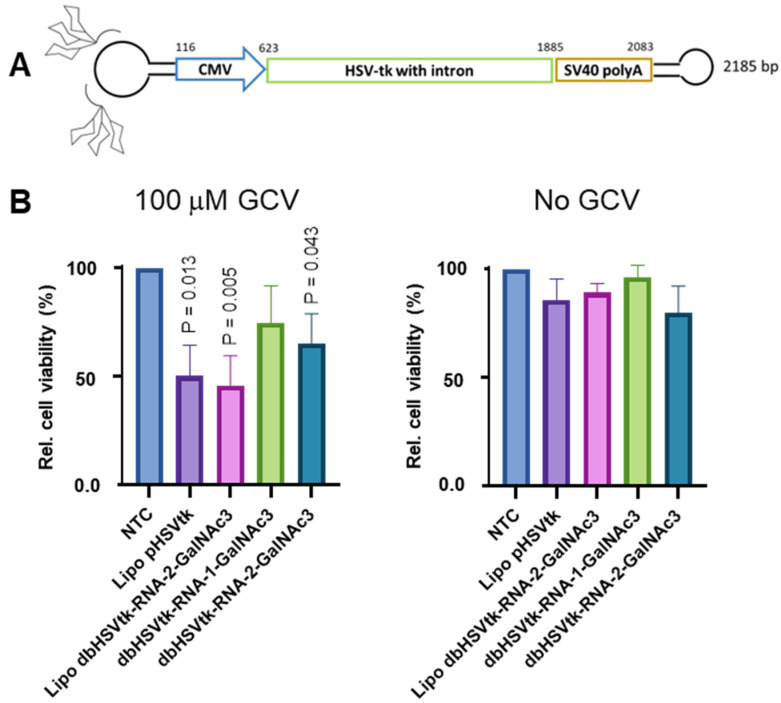
Death of HepG2 cells triggered by dbHSVtk-GalNAc3 conjugates. (**A**) Design of the HSVtk dumbbell-GalNAc3 conjugates. (**B**) Death of HepG2 cells triggered by dbHSVtk-GalNAc3 conjugates added to the cell culture medium in the presence (100 µM GCV) or absence (No GCV) of GCV monitored using the alamarBlue cell viability assay. HepG2 cells were either transfected with Lipfectamine 3000 or exposed to HSVtk expressing vectors added to the cell culture medium and cell death was monitored on day six. NTC: No transfection control; Lipo pHSVtk: HSVtk expressing plasmid delivered via lipofection; Lipo dbHSVtk-RNA-2-GalNAc3: HSVtk expressing dumbbell conjugated with 2 GalNAc3 residues via a RNA linker delivered via lipofection; dbHSVtk-RNA-1-GalNAc3: HSVtk expressing dumbbell conjugated with 1 GalNAc3 residue via a RNA linker added to the medium; dbHSVtk-RNA-2-GalNAc3: HSVtk expressing dumbbell conjugated with 2 GalNAc3 residues via a RNA linker added to the medium. Mean ± SEM (*n* = 3). Significance was tested using the unpaired student’s *t*-test.

**Figure 5 pharmaceutics-15-00370-f005:**
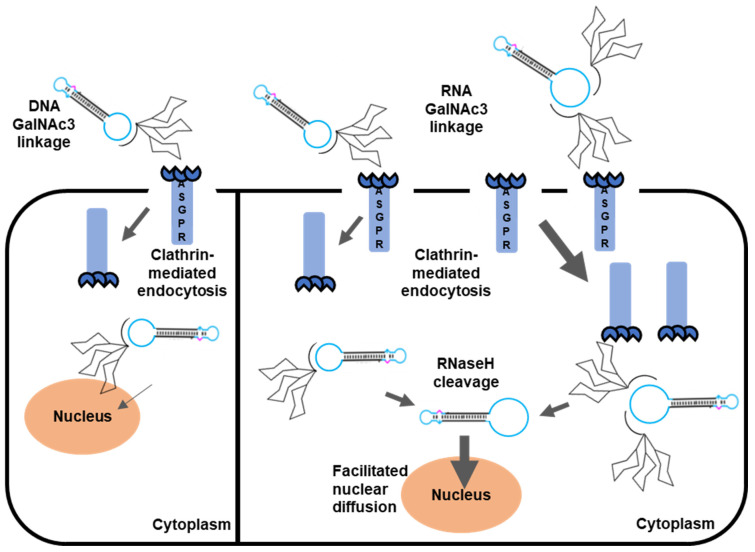
Schematic depicting the concept of GalNAc3-mediated cellular uptake and expression of dumbbell vector DNA. Single GalNAc3 conjugates bind towards one asialoglycoprotein receptor (ASGPR) and are internalized by the cell via clathrin-mediated endocytosis. Double GalNAc3 conjugates can bind to two ASGPR receptors which facilitate cellular uptake. RNA-linker- but not DNA-linker-conjugates are cleaved by the endogenous RNaseH resulting in release of the GalNAc3 residues from the dumbbell DNA. Unconjugated dumbbells are less bulky, and exhibit facilitated diffusion through the nuclear pore complex resulting in higher levels of transgene expression.

## Data Availability

Data is contained in the article or Appendix A.

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
