# Peer review of "Non-Covalent Linkage of Helper Functions to Dumbbell-Shaped DNA Vectors for Targeted Delivery"

_pharmaceutics, 2023, doi:10.3390/pharmaceutics15020370_

Round 1

Reviewer 1 Report

Manuscript "Non-covalent linkage of helper functions to dumbbell-shaped DNA vectors for targeted delivery" by Pei She Loh and Volker Patzel is devoted to delivery of DNA dumbbells with complementary oligonucleotides with a vehicle. Authors prepared complexes of GalNAc3-conjugated oligonucleotides and  a homodimer of a CD137-binding aptamer with an arm complementary to the dumbbell. Later only first type was studied, so I do not see the value for this study of the aptamer complex w/o any in vitro data. 

Figure 3A - please perform statistical analysis to confirm the difference between NTC and GalNAc derived plasmids

FIgure 3C - please provide data for GFP plasmids, perform statistical analysis and raw data used for gating. Please provide RT-qPCR data to quantify mRNA rather than pDNA that can be adsorbed on the cell or stay unproductive in endosomes.

Figure 4 - please compare results with vs w/o GCV, looks that dbHSVtk-GalNAc3 itself provide some decrease of cell viability. Please provide RT-qPCR data to quantify mRNA for HSVtk.

Presented results demonstrate only minor improvement in plasmid delivery using proposed approach. Current data do not allow to support solid conclusions presented by authors. Also HepG2 cell line is a bad model for hepatocyte delivery as an average number of ASGP receptors is usually 15-25K in comparison to freshly isolated human hepatocytes (500-1000K). 

I acknowledge the work being performed in this detailed study, but due to issues described above I suggest resubmitting the manuscript after major revision. Unfortunately, the paper cannot be accepted in a current form.

Author Response

We are very grateful for the helpful and constructive reviewer comments which we addressed as good as possible within the given time frame. 

Response to reviewer comments:

Reviewer 1:

Figure 3A: We performed the statistical analyses and added error bars to new Figure 3B which was Figure 3A in the old manuscript version.

Figure 3C: We repeated the analyses including data for the GFP plasmid and performed the statistical analyses. These data are summarised in new Figure 3E. We also added the raw data for gating in a new Supplementary Figure 3. We provided the RT-qPCR data in new Figure 3C.

Figure 4: Any apparent decrease in cell viability triggered by dbHSVtk-GalNAc3 vector is not significant. This is mentioned in the revised version of the manuscript.  

Using primary hepatocytes instead of HepG2 cells is a very good point. We ordered such cells but delivery takes time. Unfortunately, we had not enough time to generate and include such data into the revised version of the manuscript. However, as primary liver cells express much higher ASGPR levels compared with HepG2 cells, one can expect even stronger delivery and gene expression when using primary liver cells.     

Reviewer 2 Report

The article is very well written by experts in the field. However, I have some comments:

Methods:

Authors should indicate what is the "xy" site for plasmid construction.

For the sake of clarity, Authors could provide a more detailed map of db with positions of GFP and of the restriction sites evoked throughout the text.

Authors could clarify whether they used GFP (mostly UV, poor fluorescence) or EGFP.

Provider of GCV should be indicated.

Results:

Authors need to explain why they find db in NTC group (Fig. 3A).

pMaxGFP should be included as control in Fig. 3.

Authors could demonstrate exonuclease-resistance for each db construct.

For Figure 3 data, Authors need to explain why they transfected HepG2 cells in suspension as these cells are adherent. Authors need to add a graph with mean/SD and statistical analysis.

To validate hepatoma cell targeting, Authors need to perform experiments in normal liver cells.

Discussion:

Authors could indicate the expected size of db constructs.

Author Response

We are very grateful for the helpful and constructive reviewer comments which we addressed as good as possible within the given time frame. 

Response to reviewer comments:

Reviewer 2:

Methods

Cloning of the GFP plasmid template for dumbbell production was described in the Materials & Methods section of the original manuscript.

We drew detailed maps of both the GFP and HSVtk expressing dumbbells and added them as new Figures 3A and 4A in the revised version of the manuscript.

We used MaxGFP (Lonza) which was mentioned in the Material & Methods section. We now use the term MaxGFP throughout the revised version of the manuscript.

GCV was purchased from Sigma which is now mentioned in the revised version of the manuscript.

Results

The signal indicating the presence of db vector DNA in the NTC is presumably a background signal which is either due to presence of a minor contamination of the qPCR or an unspecific signal originating from the SYBR Green-based quantification. This is being discussed in the revised version of the manuscript.

The pMaxGFP control was included in the GFP flow cytometry data which are summarized in new Figure 3E of the revised manuscript version.

We included new Supplementary Figure 1 which demonstrates the exonuclease resistance of the used MaxGFP-expressing dumbbell vectors.

For the figure 3 data, we transfected HepG2 cells after trypsinization in suspension using a small media volume in order to have a higher concentration of dumbbell vector DNA and to observe stronger expression of MaxGFP. This is now mentione in the M&M section of the revised manuscript. HSVtk triggered cell death usually establishes itself as a stronger phenotype compared with GFP expression. Hence, to monitor uptake of HSVtk expression vectors, we added the HSVtk expressing dumbbells to the medium of adhered HepG2 cells. We added a graph including mean and SD.

Using normal liver cells instead of HepG2 cells is a very good point. We ordered such cells but delivery takes time. Unfortunately, we had not enough time to generate and include such data into the revised version of the manuscript. However, as primary liver cells express much higher ASGPR levels compared with HepG2 cells, one can expect even stronger delivery and gene expression when using primary liver cells.      

Discussion

The size of the GFP and HSVtk expressing dumbbells are indicated in the new Figures 3A and 4A.